# Piston-Type Optical Modulator for Dynamic Thermal Radiation Tuning Applications

**DOI:** 10.3390/ma14164372

**Published:** 2021-08-04

**Authors:** Andrew Caratenuto, Yi Zheng

**Affiliations:** 1Department of Mechanical and Industrial Engineering, Northeastern University, Boston, MA 02115, USA; caratenuto.a@northeastern.edu; 2Department of Electrical and Computer Engineering, Northeastern University, Boston, MA 02115, USA

**Keywords:** dynamic tuning, piston, actuation, blackbody cavity, environmental cooling

## Abstract

This study introduces a movable piston-like structure that provides a simple and cost-effective avenue for dynamically tuning thermal radiation. This structure leverages two materials with dissimilar optical responses—graphite and aluminum—to modulate from a state of high reflectance to a state of high absorptance. A cavity is created in the graphite to house an aluminum cylinder, which is displaced to actuate the device. In its raised state, the large aluminum surface area promotes a highly reflective response, while in its lowered state, the expanded graphite surface area and blackbody cavity-like interactions significantly enhance absorptance. By optimizing the area ratio, reflectance tunability of over 30% is achieved for nearly the entire ultraviolet, visible, and near-infrared wavelength regions. Furthermore, a theoretical analysis postulates wavelength-dependent effectivenesses as high as 0.70 for this method, indicating that tunabilities approaching 70% can be achieved by exploiting near-ideal absorbers and reflectors. The analog nature of this control method allows for an infinitely variable optical response between the upper and lower bounds of the device. These valuable characteristics would enable this material structure to serve practical applications, such as reducing cost and energy requirements for environmental temperature management operations.

## 1. Introduction

Material structures with tunable optical responses have made significant contributions through their associated engineering applications, including energy harvesting, radiative cooling, and defense [1,2,3]. Such materials can enhance existing technologies by allowing for active and passive control, lowering energy requirements, and reducing cost [1,2,4]. As a result, many dynamic tuning techniques have been studied and applied in the past years. Researchers have demonstrated control over spectral properties mechanically [3,5,6,7,8], thermally [1,2,9,10,11,12], electrically [13,14,15,16,17,18], optically [19,20,21,22], and magnetically [23,24,25,26,27,28,29], leading to a multitude of practical devices.

Each of these avenues for optical modulation can manifest using different techniques. For example, radiative thermal tuning has been achieved mechanically in the far field by applying stress in various orientations [3,6], as well as in the near field by modifying the relative displacement between two surfaces [7] with notable success. Thermal tuning has been realized by exploiting the temperature-dependent refractive indices of materials, such as silicon [10], and by modifying the liquid crystal state of a metasurface via temperature modulation [11]. Electric inputs have also been demonstrated to tune optical properties by using them as a stimulus for phase changes in materials such as vanadium dioxide [15,16], for chemical potential modification [14], and to induce a tunable optical response arising in multiple-quantum-well metasurfaces [17]. Researchers have achieved optical tuning by utilizing lasers to induce a liquid crystal phase change [21] and through ultraviolet pulse modification used to modify the resonance frequencies [22]. Additionally, magnetic fields have been applied to induce dielectric function anisotropy in both the near field [25,26] and the far field [27], to modify the orientation of nanochains within silicon dioxide to alter transmissivity [28], and to induce the piezophotonic effect via magnetic field excitation at specific frequencies [29].

This wide array of tunability mechanisms is accompanied by a wide array of applications. Tunable structures modulated via electrical, mechanical, and phase-change inputs can be employed within antennas and other communication devices to widen usable wavelength ranges and realize frequency band selectivity [4]. A large variety of phase change materials have been employed within thermal energy storage applications, due in part to their increased energy storage capabilities [30], as well as within data storage and memory applications [22]. Additionally, the modulation of spectral properties using thermal means has been shown to be suitable for optical applications, such as narrowband filters, high-sensitivity sensors, and holograms [11].

All of the aforementioned techniques realize dynamic thermal emission tuning with varying degrees of tuning effectiveness. However, the majority of these examples require microstructured or nanostructured features or nanoscale displacement gaps, typically adding substantial complexity and cost to fabrication and implementation procedures. Within this work, a straightforward but unique mechanical tuning approach is demonstrated for far-field thermal radiation that does not require such small-scale structures. Dynamic optical response control is achieved by combining two spectrally dissimilar materials—one absorber and one reflector—and modifying their relative displacements using a piston-like geometry. Thermal radiation tunability of well over 30% is realized throughout the ultraviolet (UV), visible, and near-infrared (NIR) wavelength regions by exploiting optical interactions similar to those of blackbody cavities [31]. These effects manifest within millimeter-scale representations, substantially easing the burden of complex fabrication that tunable metamaterials present, and offer infinitely variable analog tunability between the upper and lower limits of the device. The associated relationship between tunability and the top surface area ratios of each material structure is experimentally studied. The effectiveness of the method is also quantified as a function of wavelength and area ratio. These data suggest a theoretical limit of 70% tunability in the NIR region using this method with idealized bulk materials. Finally, the piston structure is expanded to include a series of individually actuatable pistons, confirming the hypotheses postulated within this work and offering even further tunability permutations. Demonstrating ample thermal radiation tunability within the UV, visible and NIR wavelength regions, this approach can be easily leveraged as a low-cost solar heat regulation device for buildings to reduce their energy consumption and environmental impact.

## 2. Materials and Methods

When light interacts with a medium, it will either be absorbed by the medium, transmitted through the medium, or reflected off of the medium. This response is heavily impacted by the dielectric function of the medium. Often, the thermal radiation response of a structure is tuned by dynamically modifying the effective dielectric functions of the medium. Instead, the piston geometry demonstrated in this work takes advantage of a simple 3D structure in order to modify the overall thermal radiation response without modifying the optical properties of the media materials. By placing a movable reflective material within a fixed absorptive material, the amount of interaction that light has with each material relative to the other can be adjusted significantly by modifying the position of the reflector.

The piston geometry consists of a base (the absorber) with a cylindrical cavity of varying diameter. A piston (the reflector) is inserted into the cavity at various displacements. When the reflector is in its raised position (flush with the top surface of the absorber), much of the incident light will bounce off this surface, thereby promoting the reflectance of the entire structure. Conversely, when the reflector is in its lowered position, additional light will enter the cavity, prompting added light interactions with the absorber that will greatly increase absorptance. These two modes are demonstrated schematically in Figure 1a, and a cross section of the sample is shown in Appendix A. This latter interaction is similar to that of a blackbody cavity, a manufacturable geometry that has been analytically validated as a promising spectrally selective solar absorber structure in past literature [31]. The cavity-like geometry of the negatively displaced piston structure results in many more reflection and absorption interactions than a simple flat surface, thereby boosting the overall absorptance. Of additional importance to this design are the scattering properties of the involved media. As the scattering efficiency of a material rises, its absorption efficiency will decrease [32]. Therefore, the absorber should ideally have a low scattering efficiency, while the reflector should have a high scattering efficiency. These properties will help to enhance the performance of this modulation method by supplementing the intended purposes of each material.

Based upon these concepts, graphite is chosen as the absorber, and aluminum is chosen as the reflector. Aluminum dowels are obtained from McMaster-Carr, and graphite ingots are purchased from OTOOLWORLD. Holes of various radii are drilled in the graphite to create a cavity, and aluminum dowels of matching sizes are inserted. Various piston displacements are achieved by lowering the aluminum dowel with respect to the top absorber surface. The piston displacement is defined as the distance from the top graphite surface to the top aluminum surface, which ranges from 0 mm to 32 mm. Within this work, the piston is actuated into position by hand using a rod. Its displacement is verified using a digital caliper. In a practical scenario, this positioning process can be controlled using an actuation device such as a piezoelectric motor. The fabricated structure cross sections are displayed in Figure 1b,c.

Reflectance measurements from 200 to 2500 nm are obtained for each displacement using the Jasco V770 spectrometer, equipped with a Jasco ISN-923 integrating sphere at an angle of 6${}^{\circ }$. This spectrometer has an accuracy of ±0.3% reflectance and is calibrated using a baseline reflectance standard before each use. To verify measurement repeatability, one sample case was measured five times, removing and replacing the sample after each measurement. These results are displayed in Appendix A and treated as representative of all other spectrometer data within this work. There are slight deviations attributable to the placement of the sample itself, amounting to a mean overall error of approximately 0.4% reflectance. Together with the spectrometer accuracy error, a total error within 0.6–1.0% reflectance can be expected for these measurements, depending upon the wavelength region. In the specific wavelength region of λ = 850 nm (±25 nm), there is a sensor change which can result in slight measurement discontinuities as high as approximately 3% for highly reflective samples. The illuminated sample area for this setup is approximately 8 mm × 8 mm. SEM images were obtained using the Supra 25 SEM at an acceleration voltage of 5 kV and analyzed using ImageJ.

The reflectance spectra of both bulk materials used are shown in Figure 2a. Transmittance is assumed to be zero as both materials are opaque in their bulk forms. In addition, the scattering and absorption properties of the graphite have been further characterized as they bear notable importance to this tuning method. In Figure 2b, the scattering, absorption, and extinction efficiencies of the graphite are plotted against the relevant wavelength range. These data are calculated based upon the Mie scattering theory [32] using the *MiePlot* software [33], which has been successfully employed to calculate various scattering properties in a number of published works [34,35,36] by using the algorithm of Bohren and Huffman [37]. In order to perform the Mie scattering calculations, the particle size and refractive index must be known. The SEM images (Figure 2c,d) show the surface structure of the graphite at various magnifications. These characterizations are used to estimate an average particle size of the graphite of 660 nm. In addition, the complex refractive index of the graphite is determined by assuming a Lorentz–Drude form of its dielectric function and using the reflectance spectrum to calculate Lorentz–Drude parameters, as is exemplified in the literature [38]. Based upon the resulting efficiencies, the somewhat strong scattering properties of the graphite are expected to have a negative impact on the absorptance of the structure when the piston is lowered. Its absorption efficiency is limited by a relatively high scattering efficiency, which may enable additional light to escape from the graphite surface. Even so, this aspect will help to demonstrate the effectiveness of the method in promoting absorption even when a non-ideal absorber material with relatively high scattering properties is employed.

Finally, to confirm the hypotheses regarding the tuning functionality of this device, a simple Monte Carlo simulation is first performed using an open-source software package [39]. Photon paths from a plane wave emission source are modeled within a 3D control volume with open boundaries, representative of the tunable piston device. A wavelength of 550 nm is selected to represent a prominent area of the solar wavelength region, and an incidence angle of 6${}^{\circ }$ is chosen to emulate the employed spectrometer. The refractive index and scattering properties of graphite are based upon the data shown in Figure 2, while the aluminum reflector is replaced with an ideal reflector to better illustrate the intended concepts.

## 3. Results

The Monte Carlo simulation results of Figure 3 verify the optical behaviors of the system described previously. With the reflective piston in its lowered position, a large amount of incident light is able to enter the cavity, as evidenced by the displayed fluence results in Figure 3b. After being reflected off of the lowered piston within the cavity, many photons are diverted into the side wall of the cavity. This prompts a significant amount of absorption in the graphite, illustrated by the many paths which penetrate this wall. In contrast, the fluence results for the raised piston (Figure 3c) show no penetration into the cavity, as expected. Some photons are still absorbed in the surrounding area of the graphite structure, but the vast majority of incident light is reflected away from the structure. These simulation results verify that raising and lowering the piston modifies the amount of interaction light experiences with each material, thereby modulating the overall optical response.

The reflectance spectra of the tunable piston-like structure for various piston displacements are shown for three piston diameter values (6.35 mm, 4.75 mm, and 3.2 mm) in Figure 4a–c, respectively. These three structures are referred to herein as Cases A, B, and C for brevity. As expected, scenarios in which the aluminum piston is flush against the top surface yield the greatest values of reflectance across the entire spectra. This orientation maximizes incident light exposure to the reflective material and minimizes both additional reflections and interactions with the absorptive graphite. The magnitude of reflection is heavily correlated with the area ratios of each case. With the area ratio defined as the ratio of actuatable top surface area to total top surface area, values of 0.50, 0.28, and 0.13 are obtained for Cases A, B, and C, respectively. In line with this concept, Case A possesses the maximum reflectances in its top position, while Case C possesses the minimum reflectance in its top position. While the reflectance of Case B in its top position is between those of A and C, its value is slightly higher than the midpoint between the top position reflectances of A and C. This is due in part to the area ratio percentage reduction between Cases A and B being smaller than that between B and C.

Conversely, when the piston is displaced below the top absorber surface, the absorption of structures of all diameters are enhanced. A maximum displacement value of 32 mm is chosen, as relatively minor absorptance gains can be anticipated at larger displacements based on the trends visible in Figure 4. Besides simply exposing more graphite surface to the incident light, this geometry also introduces additional surface reflections with the absorptive material due to the blackbody-like cavity left by the displaced piston. The effectiveness of this cavity in promoting absorption is made abundantly clear when comparing the spectra of the employed bulk graphite (Figure 2a) and the 32 mm-displaced piston of Case A (Figure 4a). The graphite maintains an absorptance of ~15% throughout the studied wavelength range, whereas the 32 mm-displaced piston structure in Case A maintains a far lower absorptance, near 5%. This phenomenon results from the additional light interactions with the graphite base brought on when the unique piston-type geometry is presented in its negatively displaced state. Even Cases B and C, which possess much smaller area ratios than Case A, boast absorptance improvements with respect to the bulk graphite when the piston is at its lowest displacement. Furthermore, this geometry reduces the high scattering effects of the graphite by allowing more light to be trapped within the cavity at any scattering angle. This enhances the tunability of this structure by widening the gap between the most reflective and most absorptive orientations, and provides significant tunability even with non-ideal materials. These factors work in concert, significantly amplifying the usefulness of this device.

Some tunable optical devices, such as many phase-change materials, are limited to on/off type control, in which only two optical behaviors can be leveraged [20,30]. On the other hand, this actuated device permits infinite variations in averaged reflectance between the upper and lower bounds. By displacing the piston less than the maximum chosen value of 32 mm, target reflectance values can be precisely selected to achieve the desired spectral response. Case A, with the largest area ratio, allows for the most consistent tunability between each step: at stepwise displacements of 8 mm ranging from 0 mm to 32 mm, reflectance is lowered by roughly 10% with each step. With smaller displacement values, even finer tuning of thermal radiation can be realized.

To further evaluate and explicitly compare the performances of the piston-type thermal emission tuning devices, the data in Figure 5 are presented. In Figure 5a, the maximum reflectance change of each structure is presented at five different wavelength values. The maximum change is based upon a modulation between the non-displaced and 32 mm-displaced states. This figure clearly shows the dramatic impact that the area ratio has on enhancing the tunability of this structure, with Case A possessing over 3 times the tunability of Case C.

Wavelength-dependent data for the reflectance of both bulk materials allow for the evaluation of optical tuning effectiveness (Figure 5b). Effectiveness ε is determined via the relation ε=Achievedtunability/Idealtunability. The intended functionality of this piston-type structure is to modulate between states of high reflection and high absorption. As such, the difference in reflectance of the reflector and absorber materials at a given wavelength is defined as the ideal tunability, while the difference in reflectance of the raised piston state (0 mm displacement) and the lowered piston state (32 mm displacement) is defined as the achieved tunability.

The resulting effectiveness values show continuity with the previously discussed trend, in that tunability effectiveness is higher for structures with larger actuatable area ratios. Because the piston diameter is directly related to the area ratio, and larger area ratios have been shown to offer increased tunability for a sample surface of constant area, the larger piston diameters result in higher effectiveness values. The structure employing a piston diameter of 6.35 mm (Case A) displays the highest effectiveness over nearly every wavelength value, with a mean value of 0.62 and a maximum value of 0.70. Effectiveness values are fairly consistent across the entire visible/NIR wavelength region, with several minor fluctuations at the peak locations of the bulk materials. Because this effectiveness is primarily based upon the area ratio and the maximum relative displacement, the effectiveness of this modulation strategy can be expected to remain consistent for geometries of larger or smaller scale if the relationship between the actuatable area and displacement is also scaled. Thus, in order to reach this high effectiveness value for scaled geometries, a maximum displacement of at least five times the piston diameter is recommended, corresponding to this ratio for Case A. This same ratio also corresponds to a displacement of 16 mm for Case C, after which smaller absorption gains are realized, further justifying this value as a useful design benchmark.

Using the reflectance difference between the bulk materials as the ideal reference allows for a definition of effectiveness that is independent of material (with the exception of scattering effects). As such, if changes in scattering effects are assumed to be negligible, tunability percentages can be estimated for alternative material choices using this metric. Therefore, the maximum theoretical tunability percentage of this structure can be determined by assuming perfect reflector and absorber materials in place of the aluminum and graphite, respectively. In this case, the Idealtunability will be equal to 100%. Thus, while the tunable percentages displayed in Figure 4 are limited to below 40%, the analyses in Figure 5b suggest that material enhancements can support a tunability as high as 70% at NIR wavelengths using a structure with an area ratio comparable to Case A.

For practical implementations, the response of this device to different angles of incidence (AOI) are important to note. Figure 5c presents the spectra results for three displacements of Case A at incidence angles ranging from 0 to 39 degrees. These measurements were performed by placing a precisely angled reflective wedge between the spectrometer window and the sample, as shown schematically in Appendix A. Each spectra is normalized with respect to the AM 1.5 solar irradiance data to present an averaged result for each angle of incidence. Because this tuning mechanism relies heavily on the multiple interactions with the internal graphite cavity, the angle of incidence is predicted to play a large role in spectral response. At higher angles of incidence and greater piston displacements, the light can be expected on average to have added impacts with the graphite cavity, thereby lowering overall reflectance. As such, absorption is increased greatly up to an AOI of about 24 degrees, and is enhanced by larger piston displacements. After 24 degrees, absorption gains are less prominent in most cases, attributable to the geometric interplay between the cavity depth and the total amount of light entering the cavity. Absorption is also enhanced at higher AOI in the 0 mm displacement case, which is partially attributed to the piston cavity being slightly larger than the piston itself, discussed further in the following sections. In addition, a very slight increase in the averaged reflectance is visible between angles of 36 and 39 degrees for the 16 mm displaced case. However, the magnitude of this change is only about 0.1% reflectance (well within the previously mentioned error bounds). As such, this constitutes an asymptotic trend for the 16 mm case, as barely any absorptance gains take place after an incidence angle of 24 degrees, illustrating a lower bound for the absorptance of Case A. Although the maximum tunability does decrease as the angle of incidence increases, the ability of this structure to modulate between a reflective response and an absorptive response is still obvious at elevated angles of incidence, helping justify the practicality of this device.

Finally, Figure 6 provides further information concerning the performance of this piston-type spectral modulation. As stated previously, the effectiveness of the actuation method is strongly correlated with the area ratio of the chosen structure. The curves in Figure 6a expand this postulation to a structure which employs four cylindrical pistons instead of one, all with diameters of 3.20 mm. The single-piston structure has an area ratio of 0.13, while the quadruple-piston structure has an area ratio of 0.50—nearly identical to that of the single-piston structure with piston diameter A. This increase in the area ratio leads to an increased reflection in the non-displaced position (by an average of 8%) and a decreased reflection in the 16 mm-displaced position (by an average of 7%). Therefore, the main impact of the packed structure with respect to the single-piston structure is enhancing tunability via both an area ratio enlargement as well additional cavity reflection interactions in the lowered position. While this result is surmisable based on the previous data, it is also valuable for confirming that the geometric feature which promotes tunability in this structure is the area ratio, rather than simply the piston diameter.

With such similar area ratios, one would expect the spectra of Case A (Figure 4a) and the 4× piston case shown in Figure 6a to be quite similar, especially in the raised and lowered extreme positions. The 32 mm-displaced piston of Case A has nearly the same reflectance as that of the 4× 16 mm-displaced orientation; while the latter is displaced only half the distance of the former, its smaller cavities present more opportunity for internal reflections which enhance absorptance. This results in an apparent balancing of absorptance performance, with both structures accomplishing nearly equal low reflectivities. Conversely, the corresponding spectra of the raised cases differ by an average of approximately 10%. This is attributed essentially to sample preparation uncertainties. The cavities which house the pistons within the graphite must be drilled slightly larger than the pistons themselves to both fit the pistons and permit displacement. This results in a small cavity around the circumference of the piston when it is not negatively displaced (i.e., in its most reflective state), which reduces reflectance. Although the area ratios of both these cases are nearly identical, the circumferential length of the 4× piston case is larger, leading to an expansion of the total area of these cavities, thereby reducing reflectance in the raised state. In a practical scenario, this effect could be diminished by more precise machining and tolerancing, as well as surface lubrication, all of which would permit piston motion more easily while minimizing the size of the circumferential cavity.

In Figure 6b, spectra for fully raised (0 mm displacement) and fully lowered (16 mm displacement) sets of four pistons are presented with respect to various alternative avenues of piston pattern actuation. Structures with one, two, and three pistons lowered are experimentally tested, with all other pistons raised. In these cases, the reflectance rises as additional pistons are raised and falls as additional pistons are lowered. This is due entirely to the modification of the top surface area ratio as pistons are actuated, offering either more reflection or more absorption area as discussed prior. Two instances are tested which raise two pistons while lowering the other two: one with two pistons in line, and another with two pistons positioned diagonal from another. Because the area ratios here are exactly the same, one would expect the associated spectra to remain identical as well. This is largely the case, with only minor variations arising which are attributable to small variations in cavity size. The curves of Figure 6b uphold the previously explained interpretations of thermal radiation tunability, again identifying the area ratio as the principal factor of importance for this modulation mechanism.

As a device capable of modulating between states of high and low absorbance, the most obvious practical application for this tunable piston-type structure is that of an adaptable building cooling device. In its lowered absorptive state, this structure can take in large amounts of incident solar energy, subsequently transferring this energy to an internal environment to aid a heating operation. On the other hand, in its raised reflective state, it can reject solar energy to reduce the heat input to an internal environment to aid a cooling operation. Both of these possibilities can greatly reduce the total energy consumption, operational costs and negative environmental impact of dwelling heating and cooling, especially given how simply the piston position can be modified.

Such an implementation has several distinct advantages that would enhance its applicability. First, in contrast with phase-change materials, which can be used to achieve a similar goal with an on/off-type selectivity [30], the piston-type cooling structure is continuously variable between its top and bottom positions. For example, the spectra at displacement positions between those measured within this work may be reliably estimated based on interpolation between the data shown herein, thereby facilitating this type of variation. This could be achieved using a small actuation device such as a piezoelectric motor. Such a strategy enables precise control of the heat flux entering the internal environment within the upper and lower bounds of the device, which would be especially useful for attaining specifically desired internal temperatures. In addition, by independently controlling different regions of pistons, the reflectance of the surface can be varied geographically, accommodating the needs of various internal building environments simultaneously. The low-cost materials and simple fabrication offer an obvious monetary advantage over the inherent fabrication complexity of many other tunable devices. Furthermore, while the manufacturing methods used within this work should ideally be improved for a practical setting to minimize losses, the scaling ability of this device is justified via the data in Figure 6. These favorable characteristics can allow the piston-type optical modulator to present notable cost and environmental benefits to temperature control operations.

## 4. Conclusions

Within this study, a unique and simplistic macroscopic optical tuning mechanism is described and experimentally evaluated. With an absorptive bulk material (graphite) and an actuatable reflective material (aluminum), the exemplified device forms a piston-like mechanism used to modify overall reflectance. At millimeter scale, this device performs well without demanding the complex fabrication methods required of micro- and nanoscale structures. Its geometry is tuned to take advantage of blackbody-type interactions, which enhance absorptance below that of bulk graphite when the piston is in its lowered position due to additional light interactions with the graphite and a reduced impact of scattering on absorption. In its raised state, the highly reflective aluminum surface blocks these interactions and greatly raises reflectance. The phenomena which enable this device to tune the optical response are verified within a Monte Carlo simulation, laying the groundwork for geometric and material optimizations through further computational studies.

Working in concert, an actuation of approximately 5× the associated diameter realizes a tunability of well over 30% across nearly the entire visible/NIR wavelength region. Furthermore, a device effectiveness is defined based on the wavelength-dependent difference in reflectance of the bulk absorber and reflector. By treating this difference as the ideal scenario, a mean wavelength-dependent effectiveness of this tunable device 0.62 is reached, along with a maximum effectiveness of 0.70 at NIR wavelengths. This high performance is obtained by optimizing the area ratio of the structure, identified as the key feature for enhancing the tunability magnitude when employing this method. Various structural factors are modified as well, enabling additional avenues for actuation and further supporting the aforementioned dependence upon the area ratio. Practical implementations which can reduce operational costs for environmental temperature control operations are discussed in detail, which would yield potential monetary and sustainability benefits. 

## Figures and Tables

**Figure 1 materials-14-04372-f001:**
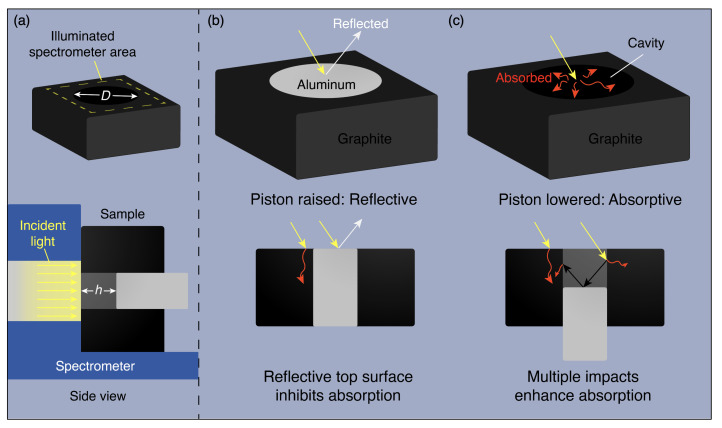
(**a**) Schematic view of spectrometer test setup, showing geometry defined by diameter *D* and displacement *h*. The side view illustrates the orientation of the spectrometer with respect to the sample. Schematic concepts of piston-like actuation for dynamic thermal radiation tuning (cross-sectional view) in (**b**) raised (reflective) and (**c**) lowered (absorptive) positions. Employing dissimilar materials and modifying the overall shape factor induces significant changes in optical response.

**Figure 2 materials-14-04372-f002:**
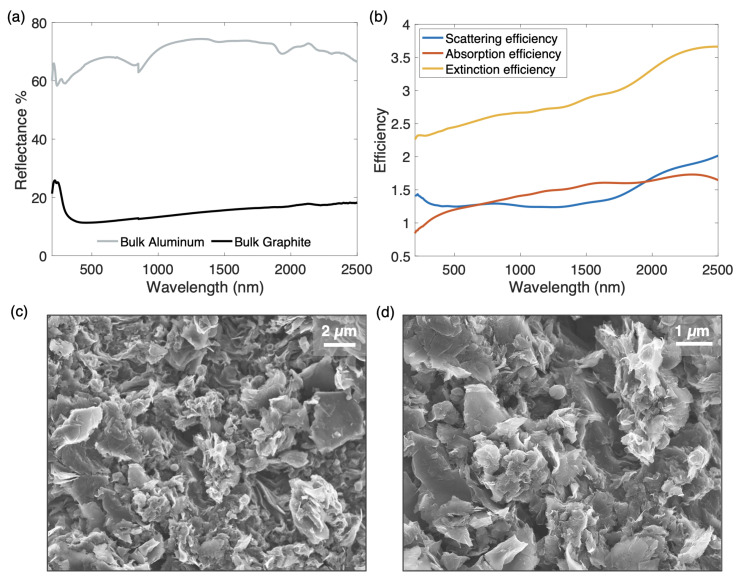
(**a**) Reflectance spectra of bulk materials used within the tunable piston structure. (**b**) Scattering, absorption, and extinction efficiencies of graphite. SEM images of graphite surface at magnifications of (**c**) 4000 and (**d**) 10,000. The SEM images are used to estimate an average graphite particle size of 660 nm.

**Figure 3 materials-14-04372-f003:**
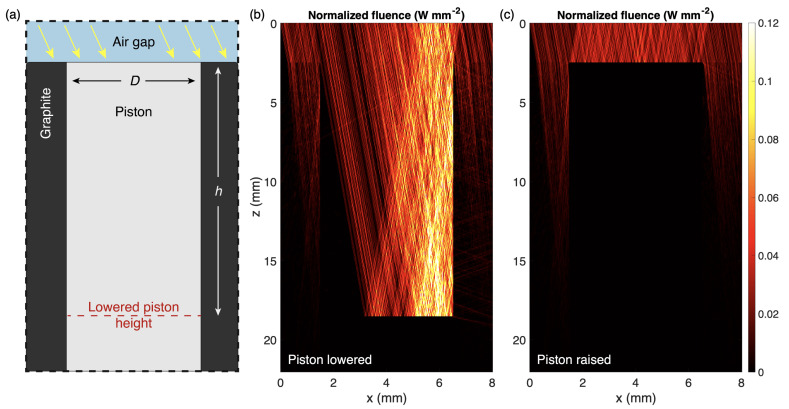
Monte Carlo simulation of tunable piston device. (**a**) Schematic of simulation control volume (cross section), illustrating the simulated geometry. The control volume measures 8 mm × 8 mm × 24 mm. The lowered piston displacement *h* is 16 mm, piston diameter *D* is 5.0 mm, and the air gap is 2.5 mm. The source is a 550 nm plane wave with an angle of incidence of 6${}^{\circ }$. Fluence results for (**b**) lowered and (**c**) raised piston structures, taken at the center cross section of the device (y = 4 mm).

**Figure 4 materials-14-04372-f004:**
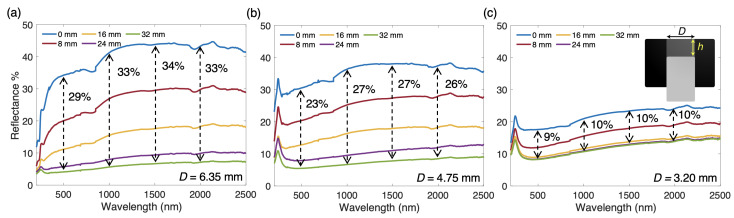
Reflectance spectra of tunable piston structures with variable piston displacement *h*, as indicated by the inset schematic. Piston diameters *D* are (**a**) 6.35 mm, (**b**) 4.75 mm, and (**c**) 3.20 mm.

**Figure 5 materials-14-04372-f005:**
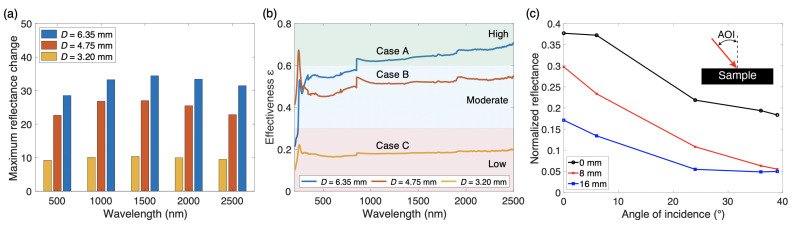
Evaluation of tunability performance. (**a**) Direct comparison of maximum reflectance change for each structure at five wavelength values. The maximum change is based upon a modulation between the non-displaced and 32 mm-displaced states. (**b**) Effectiveness of piston-type tunability with 32 mm actuation in modulating reflectance. (**c**) Spectra results for three displacements of Case A at various angles of incidence (AOI), normalized with respect to the AM 1.5 solar irradiance data to present an average result. The inset illustrates how the AOI is defined with respect to the sample.

**Figure 6 materials-14-04372-f006:**
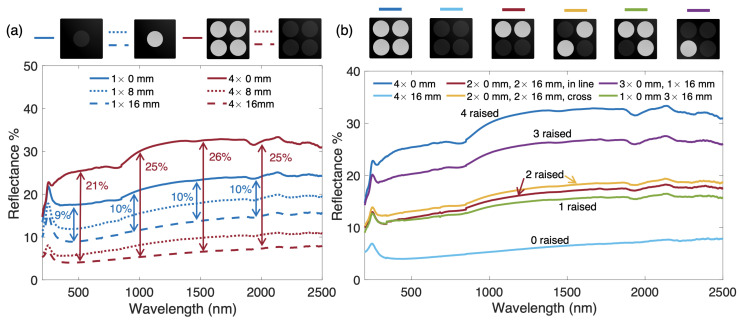
Reflectance spectra for multi-piston structures. (**a**) Spectra comparison for single piston and 4× grouped piston structures at various displacements. (**b**) Spectra comparison for 4× grouped piston structure with various combinations of piston positions. All pistons have diameter *D* = 3.20 mm.

## Data Availability

The data that support the findings of this study are available from the corresponding author upon reasonable request.

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
