# Peer review of "Piston-Type Optical Modulator for Dynamic Thermal Radiation Tuning Applications"

_materials, 2021, doi:10.3390/ma14164372_

Round 1
Reviewer 1 Report
The authors present a mechanical design to modulate the reflectance of a surface utilising movable Al pistons embedded in a Graphite block.
While the idea appears to be interesting, the manuscript is lacking scientific depth.
Major:
-- The title is misleading as there is no spectral tunability.
-- No reference spectrum of Al and graphite is shown.
-- In lines149 -- 150 the authors claim that the data suggest a tunability of up to 70% might be possible. How do the authors come to this claim?
-- Figure 2 shows three different piston diameters. One would expect that the reflectance would scale in a monotonic way with the Al area if the piston is in its top most position; however, there seems to be a jump from b) to c). This needs some discussion.
-- Furthermore, Figure 2 b) shows a non monotonic jump between piston positions while a) and c) appear to be monotonic as expected. This needs to be discussed in more detail.
-- Figure 3 b) shows discontinuities which are not physical and raise questions about the validity of the measurement. This needs at least discussion.
-- Figure 3 b) also shows that the effectiveness as a function of wavelength changes with changing piston diameter which is surprising and needs a more in depth discussion.
-- The authors claim that they have tuned the geometry to optimise the "black-body interaction". However, from the results it appears simply that a larger piston results in a better performance. Is there an optimum size?
-- The authors present results only for one incidence angle; however, for most practical application angular dependence would be of interest.
-- How representative is the variation of piston size using only one piston in a block compared to different piston sizes in different packing structures? What influence dose packing structure has?
-- Is it possible to calculate the spectrum at different piston heights as a superposition of the spectra at the extreme positions of the pistons?
-- It would be interesting to discuss if there is an optimum travel/movement range for the piston as the relative change in reflectivity decreases as the piston is lowered.
Minor:
-- Inline 108 and 224 it is not clear if the authors talk about absorbance or reflectance.
Reviewer 2 Report
I have read with interest the paper, in which the authors present a piston-type optical modulator.
The paper is clearly written and the results are interesting. Nevertheless I believe that the following aspects should be better clarified:
- The authors claim in the introduction as well as in the conclusion that a possible application is in temperature control operation. Can the authors better clarify this statement? Is the scalability of the device a fundamental property? Is it feasible?
- Can the authors comment on the tunability of this device? How is the piston translated? What is the control over this displacement? As the authors underline the importance of a tunable behavior this topic should be better discussed.
- What is the stability of the piston displacement over time?
- Would a tilted surface of the aluminum piston improve the device performances?
- As the device with better performances is the one with larger ratio (actuable surface/total surface = 0.5) why the authors did not further increase this value?
- Is the difference between achieved tunability and ideal tunability related to the actuable surface/total surface only? What are the causes that limit the achieved tunability?
- Spectral tuning is declared in the title but the behavior is quite flat over large bandwidth, can the authors clarify the title?
Reviewer 3 Report
Overall Comments
This paper describes a mechanical/optical approach for modulating some aspects of material properties. This is achieved by combining two materials, graphite and aluminium, in a piston-like format. The combination of reflectance and absorption properties of the two materials is controlled thereby achieving optical tuning/modulation.
The review of the literature is largely adequate. However, the matter of optical scattering is not considered and this is essential. Graphite can have complex scattering properties that need to be taken into account. On Page 1 a ‘theoretical analysis’ is mentioned but the paper does not appear to include such an analysis, lacking a theoretical background and analysis to explain the basic principle of the device and its performance characteristics, including further detail of the blackbody interactions referred to (pages 1, 2, 3, and 6).
The paper is submitted to the Materials Journal and, although it contains the subjects of materials science and photonics, it is not clear whether or not there is adequate content for the Materials readership. The work described also includes photonics principles and practice and so this also needs to be presented to be understood by the materials readership. I have a concern that the paper does not fulfil these requirements.
Introduction
- It would help to say more about specific applications and particular materials in which the tuning and/or modulation of actual optical properties is required .
- Line 26. Please expand on your mention of “…business sectors…”.
- Line 39; Suggest change “..external magnetic inputs..”, perhaps say ‘…magnetic fields…'
- The paragraph in Lines 27-43 lists many examples of tuning methods. You should give an idea of the success or failure/limitation of these approaches so that it becomes clear why you need to propose your piston method.
- The authors have a paper in Scientific Reports on a kind of black body system [Ref 30]. If this is relevant to the present paper then it would be useful to expand on it further.
- The main optical properties of interest includes scattering but you do not mention this nor consider the possible roles played by scatter in the various material structures alluded to.
Materials and Methods
- The description and operation of the system is not clear. The Materials and Methods section starts with a complex description of the system and its function, with fixed then moveable sections and so on. Eventually there is a conceptual drawing and description of the basic system in Figure 1 (a). There would be less confusion if Figure 1 (a) and the basic description were moved to the very beginning of the Materials and Methods section. This drawing needs improvement as does the matching description in the text and in the Figure caption.
- After the first paragraph of the Materials and Methods section, or as a separate section, it would be helpful to introduce a basic theoretical analysis of the operation of the system. This could be based on standard photon propagation principles.
- The placement of the Jasco spectrometer and associated components should be indicated in Figure 1(a) or in a separate Figure. It is essential to know this so that it becomes clear which volume of the two materials (graphite and aluminium) is being interrogated.
- The sketch in Fig 1(a) indicates a kind of photon path. Such a path could be estimated using Monte Carlo simulations, with estimates of the graphite absorption and scattering coefficients.
Results
- There is some confusion between ‘lateral displacement’ and ‘displacement’, referred to in pages 2 and 5, lines 86-87 and page 174. This needs to be clarified both in the text and in any relevant Figure. This should indicate where the ‘lateral displacements’, are actually measured.
- The Results showing reflectance spectra for different values of diameter, D, and axial displacement, h, confirm the predicted functioning of the system (Fig 2).
- The ‘tunability’ is also demonstrated (Fig 3).
- The calculated ‘area ratio’ is used appropriately, as is the ‘Effectiveness’.
- The inclusion of multiple ‘pistons’ was examined and seems feasible to achieve a higher ‘area ratio’ and therefore is useful (Fig 4).
- The paper does not appear to include a statistical analysis of the results to assess the repeatability of the measurements; precision and accuracy. How many runs were made for each of the values of h? D is fixed.
- The example application of an adaptable building cooling device is convincing.
Conclusions
- The stated conclusions are largely accurate.
- Line 225. Should not scattering be referred to alongside reflectance?
Round 2
Reviewer 1 Report
The authors have addressed all concerns and I support publication as is.
Reviewer 2 Report
The authors properly addressed the comments raised.
Author Response
Thank you for your input, we greatly appreciate your time and consideration.
Reviewer 3 Report
I agree that almost all points have been taken care of, and the paper is improved. The one area where there is still a matter to improve on, is the theoretical underpinning of the method
Previous comment: “The sketch in Fig 1(a) indicates a kind of photon path. Such a path could be estimated using Monte Carlo simulations, with estimates of the graphite absorption and scattering coefficients.”
Author’s response: “Thank you for this comment. With the well-defined geometry and the characterized graphite coefficients, such a simulation would be a valuable tool for optimizing the tunability performance of this system. However, this work aims primarily to present the general concept, describe the factors which contribute to its effectiveness, and propose and justify practical applications for this device. With the data compiled here, our group or others will be well positioned to perform these types of simulations to further improve material and geometric choices to enhance performance.”
Further comment: The authors have now included the important matter of scattering in their revised version. This is important. However, multiple photon scatter is a major determinant of photon paths and this is best studied with methods such as Monte Carlo simulation. If this were done then the fundamental description of the new device could be produced, instead of the present qualitative approach. I still feel that at least a preliminary simulation should be set out and then followed up in the next phase of the work.
Round 3
Reviewer 3 Report
I quite understand the difficult situation in which the authors find themselves, partly as they say that they do not have the expertise to carry out a Monte Carlo simulation of photon propagation. Nevertheless, their paper concerns the development and testing of a device that centres on the utilisation of optical/photonic principles and the modulation of optical parameters. The latter includes consideration of the basic optical processes of reflectance, absorption and scattering. Then, the paper inevitably discusses the paths that may be taken by photons that are launched into their device and the changes of such paths when mechanical movements of the piston relative to the stationary material are made. This stationary material (graphite) has absorptive and scattering properties that allow photon propagation modulated by the piston movement relative to the graphite.
Well, although it is quite straightforward to use this kind of qualitative description of the device and its functions we, as scientists, will require the assurance of a mathematical framework that allows us to understand and quantify the behaviour of the device; Monte Carlo simulation (MCS) is a particularly useful tool that allows us to do this.
The authors have started to investigate MCS but are concerned that to actually set up an MC simulation would take time and introduce a considerable delay in having the work done on the piston device published.
The only positive suggestion that I can make at this point is that the authors could write a paragraph or so (in the results section and a brief summary in the Conclusions) that indicates the need in future work to study photon propagation in their device and to use MCS and other modelling techniques to carry out such studies. To give you an idea of what MCS might demonstrate for you I also attach 2 pages of a chapter written by Steve Jacques, an Expert on MCS. It would be inappropriate for me to go further with this suggestion and I remain to be convinced if you do follow this advice.

Round 4
Reviewer 3 Report
The authors have now carried out a preliminary Monte Carlo Simulation and this greatly improves the scientific quality of the work. The paper is now suitable for publication.